# Redox and Nucleophilic Reactions of Naphthoquinones with Small Thiols and Their Effects on Oxidization of H_2_S to Inorganic and Organic Hydropolysulfides and Thiosulfate

**DOI:** 10.3390/ijms24087516

**Published:** 2023-04-19

**Authors:** Kenneth R. Olson, Kasey J. Clear, Yan Gao, Zhilin Ma, Nathaniel M. Cieplik, Alyssa R. Fiume, Dominic J. Gaziano, Stephen M. Kasko, Jennifer Luu, Ella Pfaff, Anthony Travlos, Cecilia Velander, Katherine J. Wilson, Elizabeth D. Edwards, Karl D. Straub, Gang Wu

**Affiliations:** 1Indiana University School of Medicine—South Bend, South Bend, IN 46617, USA; 2Department of Biological Sciences, University of Notre Dame, Notre Dame, IN 46556, USA; 3Department of Chemistry and Biochemistry, Indiana University South Bend, South Bend, IN 46615, USA; 4Central Arkansas Veteran’s Healthcare System, Little Rock, AR 72205, USA; 5Departments of Medicine and Biochemistry, University of Arkansas for Medical Sciences, Little Rock, AR 72202, USA; 6Department of Internal Medicine, The University of Texas—McGovern Medical School, Houston, TX 77030, USA

**Keywords:** reactive sulfur species, reactive oxygen species, antioxidants, naphthoquinones, naphthoquinone thiol adducts

## Abstract

Naphthoquinone (1,4-NQ) and its derivatives (NQs, juglone, plumbagin, 2-methoxy-1,4-NQ, and menadione) have a variety of therapeutic applications, many of which are attributed to redox cycling and the production of reactive oxygen species (ROS). We previously demonstrated that NQs also oxidize hydrogen sulfide (H_2_S) to reactive sulfur species (RSS), potentially conveying identical benefits. Here we use RSS-specific fluorophores, mass spectroscopy, EPR and UV-Vis spectrometry, and oxygen-sensitive optodes to examine the effects of thiols and thiol-NQ adducts on H_2_S-NQ reactions. In the presence of glutathione (GSH) and cysteine (Cys), 1,4-NQ oxidizes H_2_S to both inorganic and organic hydroper-/hydropolysulfides (R_2_S_n_, R=H, Cys, GSH; *n* = 2–4) and organic sulfoxides (GS_n_OH, *n* = 1, 2). These reactions reduce NQs and consume oxygen via a semiquinone intermediate. NQs are also reduced as they form adducts with GSH, Cys, protein thiols, and amines. Thiol, but not amine, adducts may increase or decrease H_2_S oxidation in reactions that are both NQ- and thiol-specific. Amine adducts also inhibit the formation of thiol adducts. These results suggest that NQs may react with endogenous thiols, including GSH, Cys, and protein Cys, and that these adducts may affect both thiol reactions as well as RSS production from H_2_S.

## 1. Introduction

1,4-Naphthoquinones and their derivatives (collectively termed NQs) are naturally occurring and synthetic compounds that are toxic at high levels, but at lower levels they have purported anti-inflammatory, anticancer, antibacterial, antiviral, antifungal, and antiparasitic actions as well as cytoprotective effects in essentially all organ systems [1,2,3,4,5,6,7,8,9,10,11,12]. These functions are derived in part from their ability to participate in redox cycling. As described by Kumagai et al. [13], oxidized NQs may undergo one-electron reduction to a semiquinone radical (NQ^•^) and a second one-electron reduction to hydroquinone (NQH_2_) in reactions catalyzed by cytochrome P450 reductase or other flavoprotein enzymes. Alternatively, the NQ may undergo a two-electron reduction to the fully reduced NQH_2_ in a reaction catalyzed by DT diaphorase (NADPH quinone reductase). Both reactions use NADPH as the electron donor. The semiquinone is then reoxidized (autoxidized) by molecular oxygen to the NQ and oxygen is reduced to the superoxide anion (O_2_^•−^). Hydroquinones can be reoxidized (autoxidized) by oxygen in two sequential, one-electron steps with a semiquinone intermediate or by an initial comproportionation reaction between a fully oxidized and reduced quinone that produces semiquinones that then undergo one-electron oxidation as above. Superoxide, a product of the reoxidation process, dismutes, either spontaneously or catalyzed by superoxide dismutase (SOD), to oxygen and hydrogen peroxide (H_2_O_2_). Many of the biological effects of NQ are attributed to hydrogen peroxide production.

NQs also may serve as electrophiles and form covalent adducts with nucleophiles in a Michael-type addition. Most notable are reactions with low-molecular-weight thiols, such as cysteine (Cys) and glutathione (GSH), as well as reactions with amines. Especially noteworthy are NQ-Cys adducts on regulatory proteins [13] and NQ-GSH adducts that participate in glutathionylation reactions [14,15] and direct modification of cysteines on regulatory proteins [6]. The formation of NQ-GSH adducts is also believed to deplete cells of GSH, thereby decreasing cellular resistance to oxidative stress [14,15]. However, the limited bioavailability of NQs compared to the abundance of intracellular GSH would suggest that this is not a direct effect.

We [16] recently showed that a variety of NQs catalytically oxidize H_2_S to hydroper- and hydropolysulfides, sulfite, and thiosulfate. The initial reaction appears to be either two sequential one-electron reductions of oxidized NQ to a semiquinone and then to the reduced NQ (NQH_2_) with concomitant oxidation of two H_2_S to thiyl radicals (HS^•^), the latter then combining to a persulfide (e.g., H_2_S_2_) or a single two-electron reduction of the NQ to NQH_2_ with concomitant oxidation of H_2_S to S^0^ with the subsequent reaction of S^0^ with H_2_S to make the polysulfide, H_2_S_2_. Reoxidation (autoxidation) appears to occur by either two sequential, one-electron oxidative steps or an initial comproportionation, as described above, followed by one-electron oxidation. The relative importance of either process varies with the nature of substitutions on the free 2-, 3-, and 5-carbons on the naphthoquinone backbone.

Given the ability of NQs to form adducts with thiols and amines, we hypothesize that this could affect the catalytic properties of NQ-H_2_S reactions. These would be especially relevant in intracellular environments where there can be an abundance of reactive thiols and amines. In the present study, we examine the interactions between H_2_S, GSH, Cys, propylamine (PA), and protein-Cys with NQs and their effects on NQ-catalyzed reactions. We show that reactive sulfur species (RSS) produced by NQ-catalyzed H_2_S oxidation readily react with GSH and Cys to produce a variety of GSH and Cys hydroper- and hydropolysulfides. GSH, Cys, and PA also rapidly reduce NQ as they form NQ-adducts in a process that consumes oxygen as the NQ-adducts are reoxidized. Furthermore, NQ adducts with GSH, Cys, or protein–Cys may retain, increase, or decrease their ability to oxidize H_2_S. Collectively, these studies show that there are complex interactions between small thiols that may affect intracellular redox processes through alterations in RSS metabolism. They also suggest that NQ-GSH and NQ-Cys adducts may themselves not only affect the regulatory properties of the thiols but may also affect RSS production from H_2_S in discrete intracellular compartments where they are formed.

## 2. Results

### 2.1. 1,4-Naphthoquinone Oxidizes H_2_S to Inorganic and Organic Hydroper- and Hydropolysulfides in the Presence of Low-Molecular-Weight Thiols

We have previously shown that 1,4-NQ oxidizes H_2_S to S_2_–S_6_ polysulfides [16]. Here we show that similar reactions occur in the presence of small thiols. LCMS identification of products produced by 1,4-NQ oxidation of H_2_S in the presence of Cys and GSH are shown in Figure 1. Incubation of H_2_S and 1,4-NQ with Cys produced both inorganic (H_2_S_n_, *n* = 2–5) and organic (CysS_n_H, *n* = 2–4) hydroper- and hydropolysufides; the extent of polysulfide production progressively decreased as the number of sulfur atoms increased (Figure 1A). Most hydroper- and hydropolysulfides were produced in the first 10 min and declined thereafter, except for H_2_S_2_, which remained nearly constant, and CysS_2_H, which continued to increase. Incubation with GSH (Figure 1B) produced an initial increase in H_2_S_2,3_ and GSHS_2,3_ at 10 min that declined thereafter. A small amount of GSHS_4_ and GS_2_OH was observed at 2 min and GSOH was present at 2 and 10 min. No per- or polysulfides of Cys or GSH (RS_n_R; R = Cys, GSH; *n* ≥ 2) were observed.

The TME-IAM inorganic sulfur adducts appeared to deteriorate with time as the area under the curve (AUC) for a second t = 2 min sample that was saved and analyzed after the other 2-, 10-, 30- and 60-min samples (‘**✖**’ symbol in Figure 1A,B) was often considerably lower (50% or more in some samples) compared to the first 2 min sample. The TME-IAM adducts of Cys compounds appeared fairly stable, while GSH adducts were at least 20% lower. These are points to consider when examining TME-IAM adducts over extended periods. They also suggest that the amount of hydroper- and hydropolysulfides, especially the inorganic compounds, produced in these experiments may be greater than the results indicate.

Superoxide dismutase (SOD) increases the autoxidation of NQs [17] and NQ oxidation of H_2_S [16]. The effects of superoxide dismutase (SOD) and catalase (Cat) on 1,4-NQ-mediated H_2_S oxidation in the presence of GSH were also examined. Neither enzyme, alone or together, appreciably affected H_2_S or GSH, whereas both, alone and in combination, decreased hydroper-and hydropolysulfide production (Appendix A). These results contrast with our previous studies [16], where SOD increased polysulfide production from H_2_S and 1,4-NQ (measured by SSP4 fluorescence). This could be due to the limited variety of polysulfides detected with LCMS compared to SSP4, the direct effects of GSH and Cys on H_2_S oxidation by 1,4-NQ, or the fact that SSP4 reacts quickly with polysulfides before sulfoxides are produced.

We previously used the polysulfide-specific fluorophore SSP4 to demonstrate polysulfide production by the NQ oxidation of H_2_S [16]. In preliminary experiments in the present study, it appeared that GSH interfered with this method and, indeed, this was the case. To identify possible interference by GSH and Cys, we examined the effects of these compounds on the reactions between SSP4 and the mixed polysulfide K_2_S_n_, which was found by LCMS analysis of TME-IAM derivatized K_2_S_n_ to contain 7.4 ± 1.2% H_2_S, 40.8 ± 5.9% H_2_S_2_, 41.9 ± 6.0% H_2_S_3_, 8.1 ± 1.3% H_2_S_4_, and 0.8 ± 0.2% H_2_S_5_ (mean ± SE, *n* = 7).

As shown in Appendix A, when GSH was added concurrently to SSP4 and K_2_S_n_, SSP4 fluorescence was inhibited, whereas fluorescence of SSP4 and K_2_S_n_ was unaffected when GSH was added 30 min later. Furthermore, both GSH and Cys concentration-dependently decreased fluorescence when they were added concurrently to either SSP4 and K_2_S_n_ or SSP4, K_2_S_n_, and 1,4-NQ (Appendix A). Although 1,4-NQ appeared to decrease fluorescence in Appendix A, this did not appear to be the case with K_2_S_n_, as adding 1,4-NQ immediately or 10, 30, or 60 min after adding K_2_S_n_ to SSP4 did not affect fluorescence (Appendix A). The effects of 1 mM propylamine (PA) on the SSP4 detection of 10 μM K_2_S_n_ and 1 mM GSH were also examined (Appendix A). PA essentially halved SSP4 fluorescence, whereas GSH completely inhibited fluorescence unless it was added 2 h after PA and K_2_S_n_ were mixed with SSP4. These experiments suggest that GSH and Cys directly and rapidly react with hydropolysulfides produced by the 1,4-NQ oxidation of H_2_S, whereas propylamine only partly affects the SSP4–K_2_S_n_ interaction. They also suggest that 1,4-NQ does not catalyze further reactions with K_2_S_n_.

### 2.2. Effects of GSH and Cys on Thiosulfate Production by H_2_S and 1,4-NQ

Thiosulfate is also a product of H_2_S oxidation by quinones and naphthoquinones [16,18,19]. To determine if Cys or GSH affected thiosulfate (TS) production, H_2_S was incubated with 1,4-NQ in the presence of these thiols, and thiosulfate production was measured with Ag nanoparticles (Appendix A). No TS production was detected when GSH was added and only minimal production in the presence of Cys. SOD did not increase TS production in either GSH or Cys supplemented conditions.

Additional experiments were then conducted to determine if the apparent inhibitory effects of GSH and Cys were due to these thiols reacting with some intermediate that then decreased the opportunity for thiosulfate production or if they directly interfered with the thiosulfate assay. As shown in Appendix A, when the compounds were sequentially added in the order AgNP, thiosulfate, GSH or Cys, relatively little thiosulfate was detected with GSH, whereas Cys did not appear to substantially affect thiosulfate detection. However, little thiosulfate was detected with either GSH or Cys when the AgNPs were added last (Appendix A). The inhibitory effects of GSH were also evident when different concentrations of GSH were added simultaneously to thiosulfate and AgNP, whereas they were less apparent when GSH was added ten minutes later (Appendix A). These studies show that both GSH and Cys interfere with the AgNP-thiosulfate assay, most likely by reacting directly with the AgNP.

The HPLC-MBB assay was then used to further examine the effects of GSH and Cys on thiosulfate production. As shown in Appendix A, both GSH and Cys decreased thiosulfate production by approximately 30%. This is far less of an effect than that observed with the AgNPs, although it confirms that much of the apparent inhibition seen with the thiols and AgNPs is not a direct effect on thiosulfate production, nevertheless, it does appear that both GSH and Cys have an inhibitory effect.

### 2.3. Effects of GSH and Cys on Absorbance Spectra of H_2_S Reactions with 1,4-NQ

We [16] have previously shown that oxidized 1,4-NQ has absorbance peaks at 203 nm, 246 nm, and 252 nm, a shoulder at 261 nm, and a broad peak at ~342 nm. Reduction of 1,4-NQ to 1,4-NQH_2_ with nitrogen (N_2_) produced a new peak at 240 nm, decreased the 252 peak, and blue-shifted the 342 nm peak to 333 nm (Figure 2A). Na_2_S has a peak at 228 nm and a peak at 203 nm which is a nonvolatile impurity (Figure 2B). Reduction of 1,4-NQ with H_2_S (Figure 2C) produced similar effects as reduction with N_2_, although 246 nm and 252 nm peaks were smaller, and the 342 nm peak was blue-shifted to ~305 nm. Mixed polysulfides, produced by the incubation of elemental sulfur (S_8_) with Na_2_S, have shoulders at approximately 295 nm and 376 nm (Figure 2B). GSH has a single stable peak at 203 nm (Figure 2D), and Cys has a sharp peak at 203 nm that increased with time and a shoulder at ~230 nm that initially increased then decreased over the 1576 s observation period (Figure 2E).

Time-resolved absorbance spectra of mixtures of 1,4-NQ, GSH, Cys, and H_2_S are shown in Figure 2F–M. Addition of GSH to 1,4-NQ eliminated the 246 nm peak, decreased the 252 nm peak, produced a shoulder at 256 nm and broad peaks at 307 and 417 nm (Figure 2F,G). The effects in the 246 nm and 252 nm peaks were nearly complete within the first 515 s, whereas changes in the 307 nm and 414 nm peaks developed more slowly over the 1576 s sampling period. Although GSH decreased the 246 nm and 252 nm peaks, characteristic of oxidized 1,4-NQ, the lack of a peak at 240 nm (characteristic of reduced 1,4-NQH_2_) suggests that the 1,4-NQH_2_ is rapidly reoxidized to give a spectrum of the oxidized NQ-GS adduct. Addition of H_2_S to 1,4-NQ with GSH (Figure 2H,I) produced a faster decrease in the 246 nm and 252 nm peaks, a faster increase in the 307 nm peak, and a pronounced nadir at 240 nm that was not apparent with 1,4-NQ and GSH alone. The peaks at 307 nm and 417 nm observed after the addition of GSH to 1,4-NQ were not affected by addition of H_2_S.

The addition of Cys to 1,4-NQ (Figure 2J,K) produced a transient shift in the 203 nm peak to 209 and then back to 203 nm. The 246 nm and 252 nm peaks decreased within the first 15–20 s, a shoulder appeared around 260 nm, and new peaks appeared at 304 nm and 447 nm. Over time, the 246 nm and 252 nm peaks decreased even further, the 260 nm peak became pronounced, and the 304 nm and 447 nm peaks increased. In addition to the spectrum observed with 1,4-NQ and Cys alone, adding H_2_S to 1,4-NQ and Cys produced a new peak at 226 nm and a nadir at 242 nm, and the loss of the 246 nm and 252 nm peaks occurred within the first few seconds after adding H_2_S (Figure 2L,M).

In general, the effects of GSH and Cys on the 1,4-NQ spectrum were similar. Both produced new peaks at 256 nm, 307 nm, and 417 nm (GSH) and 260 nm, 304 nm, and 447 nm (Cys), although Cys appeared to be more effective than GSH in reducing 1,4-NQ based on faster and more pronounced spectral shifts with Cys than with GSH. H_2_S also appeared to produce similar effects when added to GSH/1,4-NQ and Cys/1,4-NQ, especially at 226/7 nm and 242 nm. The pronounced peaks produced by GSH and Cys at 417 nm and 457 nm, respectively, which were not seen with either oxidized 1,4-NQ or reduced 1,4-NQH_2_, are suggestive of one or more new molecules, most likely sulfur adducts and their semiquinones. This was confirmed with EPR and LCMS analysis (cf. Section 2.5.1 and Section 2.5.2).

### 2.4. H_2_S Reaction with Specific Naphthoquinones Consumes O_2_ and Is Variously Affected by GSH and Cys

We have previously shown that *para*-quinones and NQs catalytically oxidize H_2_S to polysulfides and consume oxygen in the process [16,18,19]. The measurement of oxygen consumption also provides an alternative approach to fluorophores for examining the effects of GSH and Cys on the H_2_S/NQ reactions.

As shown in Figure 3A, O_2_ consumption increased when 1,4-NQ was added to H_2_S and this was further increased by the addition of either Cys or GSH, with Cys being more efficacious. This effect was not observed with either cystine or methionine, suggesting that a free SH or S^−^ is required. GSH and Cys also concentration-dependently increased O_2_ consumption by 1,4-NQ and H_2_S (Figure 3B–D). The maximal rate of O_2_ consumption by either 10 or 30 μM 1,4-NQ and GSH or with 30 μΜ 1,4-NQ and Cys appeared to occur with GSH/Cys concentrations between 30 and 100 μM. The effects of GSH and Cys on O_2_ consumption by H_2_S and juglone were essentially similar to 1,4-NQ (Figure 3E).

The effects of GSH and Cys on O_2_ consumption by H_2_S and other NQs were more variable (Figure 3F–J). O_2_ consumption by lawsone and H_2_S was minimal and only slightly affected by either GSH or Cys (Figure 3F). GSH decreased O_2_ consumption by 2-MNQ (Figure 3G), increased it with menadione (Figure 3H), and, after a delay, increased it with plumbagin (Figure 3I,J). Cys greatly increased O_2_ consumption by H_2_S and 2-MNQ together and had no effect with menadione or plumbagin under the same conditions. Collectively, these results indicate that substitutions on the parent 1,4-NQ profoundly and specifically affect the effects of GSH and Cys on O_2_ consumption. Given the fact that the concentration of these thiols was at least 30-fold greater than the NQs, it is likely that the effects are mediated through S-adducts on the NQ rather than thiol sequestration or other reactions with polysulfides.

### 2.5. Formation of 1,4-NQ Thiol S-Adducts and Their Effects on H_2_S Oxidation

Naphthoquinones with an open position on the 2 or 3 carbon, adjacent to the carbonyls, may react with thiol or amine nucleophiles and form adducts via a 1,4 reductive Michael addition reaction [14,17]. Importantly, S-adducts with low-molecular-weight (LMW) thiols, such as GSH and Cys, as well as with protein thiolates, have been suggested to affect intracellular antioxidant mechanisms [14,15], but the expected low intracellular concentration of exogenously administered NQs relative to GSH argue against a direct effect. S-adducts may also directly affect protein function, e.g., Keap-1 [6]. The effects of these adducts on H_2_S oxidation are not known and given the relatively high intracellular concentrations of GSH and protein thiolates, it is important to determine if these adducts affect H_2_S oxidation and thereby offer dual functionality. These questions are examined in the following sections.

#### 2.5.1. EPR Examination of GSH, Cys and H_2_S Reactions with 1,4-NQ

EPR spectra of 1,4-NQ semiquinone and reactions between oxidized 1,4-NQ and increasing concentrations of GSH, Cys, propylamine, and H_2_S are shown in Figure 4A–E. There is no noticeable spectrum of oxidized 1,4-NQ dissolved in either 21% oxygen or 0% oxygen, but if reduced to the hydroquinone (1,4-NQH_2_) with NaBH_4_ (0.15:1. NaBH_4_:1,4-NQ), a 1,4-NQ semiquinone becomes evident (Figure 4A). This likely is the result of a comproportionation reaction between the newly reduced 1,4-NQH_2_ and the remaining oxidized 1,4-NQ. Addition of GSH to 1,4-NQ produced a somewhat similar semiquinone spectrum at a low GSH:NQ ratio; however, this disappeared as the GSH:NQ ratio approached 0.5:1 (Figure 4B). Similar EPR spectra were observed for the reactions between Cys and 1,4-NQ, again with the loss of signal as the Cys:NQ ratio approached 0.5:1 (Figure 4C). Although these reactions were carried out in buffer equilibrated with room air, it is likely that the solution quickly became hypoxic as even 600 μM thiol (0.15:1 thiol:1,4-NQ ratio) reacting with the NQ will quickly consume all the oxygen (~200 μM, see Section 2.5.4 below). The EPR spectrum of 0.25:1 propylamine was identical to that of reduced 1,4-NQ. The spectrum was first observed at a propylamine:1,4-NQ ratio of 0.5:1 and persisted to the highest ratio (10:1) examined (Figure 4D).

An EPR spectrum first appeared at an H_2_S:1,4-NQ ratio of 0.15:1 (Figure 4E), and it was identical to that produced by the 1,4-NQ semiquinone (Figure 4A), including the hyperfine splitting from the two equivalent protons 2 and 3 and the four equivalent protons 5, 6, 7, and 8. This spectrum quickly changed to a different pattern at a H_2_S:1,4-NQ ratio of 0.25:1, with a quintet overlapped with other unresolved EPR features. The single quintet suggests a radical species with hyperfine splitting from only four protons. This radical species was likely due to the product from the Michael addition reaction of HS^−^ to 1,4-NQ, substituting the protons on carbons 2 and 3 with SH groups. The oxidized 1,4-NQ and reduced 1,4-NQ-2SH adduct (2,3-SH-1,4NQ) then comproportionated to generate the semiquinone radicals of 1,4-NQ and 2,3-SH-1,4-NQ. At higher H_2_S:1,4-NQ ratios (0.25:1 and above), the yield of radical species increased significantly, showing more hyperfine splittings that are distinct from that of the 1,4-NQ semiquinone radical (Figure 4E). The complicated EPR signatures are likely due to multiple radical species, probably resulting from the further substitution of protons at carbons 5, 6, 7 or 8 by excess HS^−^. Repeating the above experiment with 1,4-NQ in buffer that was not sparged with N_2_ produced essentially similar results, albeit with an improved signal suggesting the contribution of autoxidation. Collectively, these results indicate the H_2_S produces semiquinone radicals unique to this thiol.

Different mixtures of 1,4-NQ and either GSH or Cys were examined to further characterize these reactions Appendix A, respectively). The EPR spectrum was typically silent after the addition of equal parts of 4 mM GSH and 4 mM 1,4-NQ (1:1 GSH:1,4-NQ ratio, Appendix A), whereas a radical was produced after diluting the reaction solution with 4 mM 1,4-NQ (final GSH:1,4-NQ ratio = 0.25:1, Appendix A). The spectrum was similar to that of 1,4-NQ semiquinone (cf. Figure 4C) but with some unresolved features overlapped with the central quintet, suggesting a mixture of radicals dominating 1,4-NQ semiquinone and, to a lesser extent, other radical(s), probably 1,4-NQ-GSH semiquinone.

The EPR spectrum with 1,4-NQ and Cys (Appendix A) was identical to that of the 1,4-NQ semiquinone. Interestingly, the semiquinone signal could be turned off or on depending on the Cys:1,4-NQ ratio. The spectrum was silent with a 1:1 Cys:1,4-NQ ratio (Appendix A); the spectrum appeared if the reaction solution was mixed with 1,4-NQ, dropping the Cys:1,4-NQ ratio to 0.3:1 (Appendix A). On the other hand, if the Cys:1,4-NQ ratio was decreased only to 0.6:1 by mixing 1:1 Cys:1,4-NQ reaction solution with less 1,4-NQ, only small EPR signals were observed (Appendix A). When this solution was further diluted with 1,4-NQ to a 0.3:1 Cys:1,4-NQ ratio, the strong signal of the 1,4-NQ semiquinone reappeared (Appendix A). This supports observations that the 1,4-NQ-Cys semiquinone is sensitive to the 1,4-NQ:Cys ratio and that a 0.5:1 ratio is the transition point. These results are suggestive of a reversible comproportionation reaction being the principal pathway for the formation of the semiquinones. This also implies that the comproportionation reaction is slower than the reduction of NQ by the thiols.

#### 2.5.2. LCMS Identification of 1,4-NQ-GSH and 1,4-NQ-Cys Adducts

Incubation of GSH with 1,4-NQ produced nearly equal amounts of 1,4-NQ-GSH and 1,4-NQH_2_-GSH. With 250 μM GSH and 1 mM 1,4-NQ (0.25× GSH), 60.2% and 36.4% of the total NQ-GSH compounds detected were 1,4-NQ-GSH and 1,4-NQH_2_-GSH, respectively. With 500 μM GSH (0.5× GSH), there were 44.3% and 49.2% 1,4-NQ-GSH and 1,4-NQH_2_-GSH, and at a 1:1 GSH:NQ ratio, 41.1% and 47.5% were 1,4-NQ-GSH and 1,4-NQH_2_-GSH (all *n* = 2; Figure 4F). Different adducts were produced by incubation of Cys with 1,4-NQ (Figure 4G). With 250 μM Cys and 1 mM 1,4-NQ (0.25× Cys), 94.2% was 1,4-NQ-Cys and 3% was 1,4-NQ-2Cys. As the Cys:NQ ratio increased progressively more 2Cys adducts were produced. With 500 μM Cys (0.5× Cys), there was 68% 1,4-NQ-Cys and 23% 1,4-NQ-2Cys, and with 1 mM Cys (1× Cys), 82% of the total was 1,4-NQ-2Cys and 11.5% was 1,4-NQH_2_-2Cys (all *n* = 2). These results show that a single GSH adduct is favored on carbon 2 or 3, which likely imposes steric hindrance preventing further GSH substitution on the other 2 or 3 carbon. Conversely, 2Cys may become predominant at higher Cys:NQ ratios as less steric effects are expected. They also suggest that the reduced NQ-thiol adducts are autoxidized more readily than their unsubstituted parent compounds, which is consistent with previous reports [20].

#### 2.5.3. EPR Examination of GSH, Cys and H_2_S Reactions with Other NQs

The EPR spectra of reactions of juglone, plumbagin, and 2-MNQ with GSH, Cys, and H_2_S are shown in Appendix A, respectively.

Reduction of juglone with NaBH_4_ (0.25:1 NaBH_4_:juglone ratio) produced a spectrum of a juglone semiquinone radical (Appendix A). Similar EPR signatures were observed in reactions of juglone with low levels of GSH or Cys (Appendix A). The EPR intensity decreased in the reactions with increasing GSH and disappeared when the GSH:juglone ratio reached 1:1 when the experiment was performed in room-air equilibrated buffer, whereas it disappeared when the ratio reached 0.5:1 when the experiment was performed in anoxic conditions. The EPR signatures were also significantly different from that of the juglone semiquinone radical, likely due to multiple radical species. In the reaction with Cys, the EPR signal disappeared when the Cys:juglone ratio was ~0.25:1 and above. Addition of H_2_S to juglone produced an EPR spectrum that was essentially identical to that of the juglone semiquinone radical at a H_2_S:juglone ratio of ~0.15:1; at higher H_2_S:juglone ratios this EPR lineshape changed significantly (Appendix A). The strongest signal was observed at a 2:1 H_2_S:juglone ratio and decreased thereafter.

The EPR spectrum of plumbagin reduced with NaBH_4_ (0.25:1 NaBH_4_: plumbagin ratio) showed a linewidth significantly broader than those observed for the reduction of juglone with NaBH_4_. The reason is unclear but may be related to the 2-methyl group on plumbagin (Appendix A). The EPR spectra of the reactions of plumbagin with different ratios of GSH in either room air or anoxia were overall similar and broader compared to those observed in the reactions of GSH with juglone (Appendix A). Reactions of plumbagin with low ratios of H_2_S (H_2_S:plumbagin ~0.15:1–0.25:1) produced spectra that were identical to that of the NaBH_4_-plumbagin spectrum but became more complicated with higher intensities and a narrower linewidth as the ratio of H_2_S:plumbagin was increased, the signal size reaching a maximum at a 1:1 H_2_S:plumbagin ratio and declining thereafter.

Reduction of 2-MNQ with NaBH_4_ produced a distinct spectrum; however, unlike either 1,4-NQ, plumbagin, or juglone, no EPR signal was observed in the reaction of 2-MNQ with either GSH or Cys (Appendix A). Conversely, the addition of H_2_S to 2-MNQ produced a spectrum that was essentially identical to that produced by NaBH_4_, and the intensity appeared to increase as the H_2_S:2-MNQ concentration ratio increased from 0.05:1 to 5.0:1 (Appendix A).

Collectively, these results suggest that at low concentrations H_2_S reduced juglone to the NQH_2_, but as the H_2_S concentration was increased a different, or several different NQ radicals, possibly due to SH-substituted quinones, were formed. Low concentrations of H_2_S also reduced plumbagin to the NQH_2_, while at high concentrations H_2_S generated plumbagin semiquinone radicals as well as other NQ radicals, which were likely due to SH-substituted plumbagin quinones. However, it should be noted that all the g values were typical of an oxygen-centered radical, not a sulfur-centered radical.

#### 2.5.4. Oxygen Consumption during Formation of GSH and Cys Adducts

We next examined oxygen consumption during reactions between 1,4-NQ and GSH or Cys to determine if the 1,4-NQ semiquinone, detected by EPR (Figure 4), was the result of O_2_-dependent autoxidation, comproportionation, or both. Oxygen was rapidly consumed when either 200 μM GSH or 280 μM Cys was added to 4 mM 1,4-NQ (Figure 5A). This ratio, 0.05:1 for GSH:1,4-NQ and 0.07:1 for Cys:1,4-NQ, is the same thiol:NQ ratio that produced a strong EPR signal (Figure 4). Progressively higher thiol:1,4-NQ ratios consistently and rapidly consumed oxygen, which suggests that while oxygen may be involved in the initial reaction(s) between the thiols and 1,4-NQ, subsequent reactions occurred in anoxia or extreme hypoxia. It is also likely that oxygen becomes progressively less relevant in the EPR spectrum as the thiol:NQ ratio increases. The rapid decline in oxygen tension that was consistently observed also suggests that adduct formation is a rapid reaction followed by rapid autoxidation of either the reduced naphthoquinone-S adduct (NQH_2_-S) or a reduced NQ that was produced by a reaction between NQH_2_-S and oxidized NQ.

To further examine oxygen consumption by thiol-NQ reactions, we used lower concentrations of thiols and 1,4-NQ that we predicted would not consume all of the oxygen. As shown in Figure 5B–E, incremental additions of thiols to 100 μM 1,4-NQ or incremental additions of 1,4-NQ to 100 μM GSH or Cys produced essentially identical concentration-dependent increases in oxygen consumption, albeit slightly more oxygen was consumed with Cys-NQ than with GSH-NQ. Furthermore, the amount of oxygen consumed was essentially equivalent to the amount of thiol or 1,4-NQ that was sequentially added. This suggests that the amount of oxygen consumed in these reactions is stoichiometrically related to and limited by the net amount of 1,4-NQ-thiol produced and not a catalytic process.

Somewhat unexpectedly, the amount of oxygen consumed by consecutive additions of 25 μM thiol to 100 μM 1,4-NQ progressively decreased such that there was little oxygen consumed after the second addition of GSH or the third addition of Cys (Figure 5F–I). However, the cumulative amount of oxygen consumed by serial additions of 25 μM GSH was similar to that produced by a single 125 μM GSH bolus when corrected for the total elapsed time. This suggests that reactions with these low concentrations of compounds are somewhat offset by continuous oxygen diffusion into the reaction chamber, which is evident from the increase in oxygen tension within 10–15 min after the compounds are mixed. Regardless, it is evident that addition of either GSH or Cys reduces 1,4-NQ and is rapidly reoxidized by molecular oxygen.

The observation that the formation of 1,4-NQ-thiol adducts consumed oxygen prompted further investigation into other NQ-thiol adducts (Appendix A). Oxygen consumption by the addition of 300 μM GSH to 30 μM NQs progressively decreased in the order 1,4-NQ~juglone > plumbagin > menadione > 2-MNQ with essentially no oxygen consumed when GSH was added to 2-MNQ. Superoxide dismutase did not appreciably affect oxygen consumption in any of these reactions. Oxygen consumption by the addition of Cys under similar conditions was greater than that for the reaction of GSH with the respective NQ and decreased from 1,4-NQ to 2-MNQ, again with little oxygen being consumed by the latter. Superoxide dismutase decreased oxygen consumption in the reaction of 1,4-NQ with Cys, but it did not appreciably affect oxygen consumption in any of the other reactions. These results show that (1) the effects of the formation of thiol adducts on oxygen consumption decrease with an increase in the complexity of the side groups on the quinoid ring, (2) more oxygen is consumed, and at a faster rate, in the formation of Cys adducts compared to GSH adducts as there is likely less steric hindrance, and (3) with the exception of the reaction of 1,4-NQ with Cys, SOD has little effect on the formation of these adducts.

#### 2.5.5. Oxygen Consumption during Formation of NQ-Propylamine Adducts

N-propylamine (PA) reportedly reacts in 5 min with 1,4-NQ in water at room temperature and physiological pH to form 1,4-NQ-PA adducts with 95% yield [21]. Several experiments were performed to determine if PA reactions with 1,4-NQ also consumed oxygen and if NQ-PA adducts affected H_2_S oxidation. As shown in Appendix A, adding 1 mM 1,4-NQ to 1 mM PA did not affect oxygen consumption, whereas oxygen consumption progressively increased when 2.5 mM and 5 mM PA were added to 1 mM 1,4-NQ. Adding 150 μM 1,4-NQ to 5 mM PA also increased oxygen consumption suggesting that both the 1,4-NQ and PA concentrations are important. This was confirmed by consecutive additions of 1,4-NQ to 5 mM PA that produced a concentration-dependent increase in oxygen consumption (Appendix A). These results also show that oxygen is consumed when a 1,4-NQ-PA adduct is formed. However, compared to other reductants, i.e., GSH or Cys (Figure 5), or dithiothreitol or ascorbic acid [16], higher PA concentrations are needed to react with 1,4-NQ and consume oxygen, even 1 mM 1,4-NQ and 5 mM PA only consumed approximately 160 μM of oxygen, which is far less than that consumed by 1,4-NQ with either GSH or Cys (Figure 5).

To determine if 1,4-NQ-PA adducts affected oxygen consumption when GSH was subsequently added, we initially incubated 100 μM 1,4-NQ with 0, 100, or 500 μM PA for 15 min and then added 100 μM GSH or Cys and observed that PA did not affect the oxygen consumed during the formation of a 1,4-NQ-GSH or 1,4-NQ-Cys adduct (Appendix A). We then incubated 5 mM 1,4-NQ with 5 mM PA for 3 h and then diluted the 1,4-NQ-PA adduct to 100 μM, 500 μM, or 1.38 mM and measured the oxygen consumption when GSH was added in an equimolar ratio, i.e., 100:100 μM, 500:500 μM, or 1.38:1.38 mM GSH:1,4-NQ-PA ratio. As shown in Appendix A, oxygen consumption was only slightly increased when GSH was added to 1,4-NQ, even when 1.38 mM GSH was added to 1.38 mM 1,4-NQ-PA adducts. Collectively, these results suggest that the formation of the 1,4-NQ-PA adduct is slower than that reported by Yadav et al. [21], but once established, 1,4-NQ-PA adducts effectively prevent the formation of thiol adducts.

The effects of propylamine on H_2_S oxidation by 1,4-NQ are shown in Appendix A. Propylamine did not affect oxygen consumption when added to H_2_S, 1,4-NQ plus H_2_S, or H_2_S with 1,4-NQ and GSH. These results show that once a 1,4-NQ-PA adduct is formed, it is relatively impervious to subsequent attack by GSH. However, PA did not affect the GSH-stimulated increase in H_2_S oxidation. This suggests that either the PA was displaced by GSH or that GSH increased oxygen consumption by removing inorganic persulfide products from the 1,4-NQ-catalyzed oxidation of H_2_S, and/or by forming GSH sulfoxides. It also suggests that if PA tightly occupies both the 2 and 3 carbons on 1,4-NQ, oxidation of H_2_S occurs at the carbonyl groups.

#### 2.5.6. H_2_S Oxidation and Polysulfide Production by NQ-Thiol, NQ-Propylamine and NQ-H_2_S Adducts

It was clear from our preliminary studies that elevated concentrations of GSH and Cys inhibited the SSP4 detection of polysulfides (Appendix A). To overcome this, we incubated high concentrations (1 mM) of NQs with similar concentrations of GSH, Cys, PA, and H_2_S for one hour to allow the adducts to form. The adducts were then diluted to a final NQ concentration of 10 μM or 30 μM. By incubating equimolar concentrations of NQ with thiols or PA, we assumed that much of the free thiol and PA would be consumed in forming the adducts and the concentration of the remaining unreacted compounds would be further reduced by the 1/100 or 1/30 dilution.

The effects of NQ-sulfur and NQ-amine adducts on H_2_S oxidation as measured by polysulfide production are shown in Figure 6 and Appendix A. Polysulfide production (SSP4 fluorescence) by 1,4-NQ-GSH and 1,4-NQ-Cys adducts was significantly greater, nearly twice as much in most instances, than that produced by 1,4-NQ alone. Overall, more polysulfides were produced by the GSH adducts. 1,4-NQ-PA or 1,4-NQ-H_2_S adducts only marginally and inconsistently affected polysulfide production (Figure 6).

Polysulfide production was also doubled by juglone-GSH adducts; however, it was not affected by juglone-Cys or juglone-PA adducts, with the exception of a potent inhibitory effect when 30 μM juglone-PA adduct was incubated with 300 μM H_2_S (Appendix A).

Polysulfide production was increased four-fold by 10 μM plumbagin-GSH adducts but was not affected by 30 μM plumbagin-GSH adducts (Appendix A). Plumbagin-Cys adducts did not affect polysulfide production at 10 μM plumbagin-Cys but inhibited production at 30 μM. Propylamine slightly increased polysulfide production and both 100 μM and 300 μM H_2_S more than doubled polysulfide production at 10 μM plumbagin-H_2_S but had no effect at 30 μM plumbagin-H_2_S.

Menadione-GSH adducts produced nearly ten-fold increases in polysulfide production; Cys adducts doubled polysulfide production at 10 μM menadione-Cys but did not affect it at 30 μM (Appendix A). Menadione-H_2_S adducts increased polysulfide production by two- to three-fold, whereas PA did not have any noticeable effect.

2-MNQ adducts were noticeably different, the GSH adducts did not affect polysulfide production, whereas polysulfide production was tripled by the Cys adducts (Appendix A). Propylamine adducts did not affect polysulfide production and only the 30 μM 2-MNQ-H_2_S increased polysulfides.

Since we were not sure that we had completely removed GSH or Cys from the above reactions, we then incubated 10 μM NQs with 300 μM H_2_S and then added 10 μM of either GSH or Cys to determine if these low residual concentrations of thiol affected the SSP4 detection of polysulfides. As shown in Appendix A, the effects of 10 μM GSH on polysulfide production were relatively small compared to the effects observed when the NQs were pre-incubated with GSH (Figure 6 and Appendix A). Adding 10 μM Cys directly to 1,4-NQ (Appendix A) had a more pronounced effect on polysulfide production than adding 10 μM GSH to 1,4-NQ (Appendix A), but other than that, adding 10 μM Cys directly to other NQs had only a minimal effect on polysulfide production. It is also clear from these figures that the onset of polysulfide production was the most rapid for juglone followed by 1,4-NQ and then plumbagin and even later for menadione and 2-MNQ. These observations correlate with our previous observations [16] on the propensity for juglone and 1,4-NQ to rapidly reoxidize after an initial comproportionation reaction, while the others with more substitutions on the quinoid ring depend more on an initial oxidation by oxygen, which is a slower and less favorable process. The slow but sustained H_2_S oxidation by menadione and 2-MNQ was confirmed by following this reaction for 4 h (Appendix A).

#### 2.5.7. Effects of Thiol:NQ Ratio on Polysulfide Production from H_2_S

As NQ-thiol adducts clearly affect H_2_S oxidation, we examined the effects of the ratio of thiol to NQ on polysulfide production. One mM of NQ was incubated with various concentrations of thiols for 1 h in an open container to autoxidize the NQ-thiol adduct, then it was diluted to 10 μM NQ and incubated with 300 μM H_2_S and SSP4 in a tape-covered well-plate for 90 min. As shown in Figure 7A, GSH concentration-dependently increased polysulfide production for all NQs except 2-MNQ. Plumbagin-GSH adducts appeared to be the most efficacious; 0.063 mM GSH per mM PB doubled polysulfide production and 1 mM GSH:1 mM PB produced more polysulfides than any other NQ-thiol combination. The general order of sensitivity was PB > Mdn > 1,4-NQ~Jug. Polysulfide production by 2-MNQ was slightly increased when incubated with 0.25 mM GSH but decreased at progressively higher GSH concentrations; however, these reactions were only monitored for 90 min and the effects of menadione and 2-MNQ could have been underestimated.

Except for 2-MNQ (and to a lesser extent, menadione), the effects of Cys were less concentration dependent (Figure 7B). Polysulfide production was increased at a 0.063:1 mM Cys:NQ for both 2-MNQ- and Mdn-Cys adducts and these effects at 90 min were also likely to be underestimated.

#### 2.5.8. Potential Reactivity of 1,4-NQ-Protein Adducts

1,4-NQ forms adducts with the reactive cysteines on proteins, including bovine serum albumin (BSA) and hemoglobin [22]. These observations were confirmed in UV-Vis absorbance (Appendix A) and EPR (Appendix A) studies.

Addition of BSA to 1,4-NQ decreased absorbance of the 252 nm peak, increased absorbance of the 260 nm and 265 nm peaks, and produced a broad peak at ~407 nm. The decrease in absorbance at 252 nm and increases at 260 nm and 407 nm were consistent with the reduction of 1,4-NQ by Cys (Figure 2F,G) and suggest that the 1,4-NQ-BSA adduct is on the reactive β93 Cys of BSA. Addition of BSA to 1,4-NQ produced an EPR spectrum that was identical to reduction of 1,4-NQ with NaBH_4_ and reaction with Cys at low Cys:1,4-NQ concentration ratio (Figure 4A,C). An identical spectrum was observed when 1,4-NQ was reacted with hemoglobin. These experiments suggest that 1,4-NQ-protein thiol adducts retain their ability to redox cycle.

We initially tried to determine if the 1,4-NQ bovine serum albumin (BSA) adduct (1,4-NQ-BSA) could oxidize H_2_S to polysulfides using SSP4; however, BSA concentration-dependently inhibited the SSP4 reaction with polysulfides (Appendix A). Parenthetically, it is also noteworthy that the SSP4/polysulfide (K_2_S_n_) reaction is approximately 75% inhibited over the range of plasma BSA (55–75 μM), which may call into question the use of SSP4 to measure polysulfides in the presence of proteins. We then examined the effects of the 1,4-NQ-BSA adduct on H_2_S consumption as BSA did not appear to substantially interfere with the H_2_S/AzMC reaction (Appendix A). As shown in Appendix A, 1,4-NQ consumed approximately 40% of the H_2_S in 30 min and the 1,4-NQ-BSA adduct consumed 30% of the H_2_S. These results show that the 1,4-NQ-BSA adduct retains the ability to oxidize H_2_S, albeit with a slight reduction in efficiency. Interestingly, and for reasons unknown, BSA also appeared to slightly increase fluorescence from the AzMC/H_2_S reaction (Appendix A), but neither BSA alone nor BSA added to 1,4-NQ affected oxygen consumption (Appendix A).

Collectively, these results show that NQ-thiol adducts can affect the catalytic efficiency of NQs to oxidize H_2_S. This appears to be specific for different thiols, for the type and location of substitutions on the NQ benzene and quinoid rings, and for the relative abundance of GSH, Cys, and reactive protein Cys.

## 3. Discussion

Naphthoquinones are well-known for their ability to participate in redox cycling and to function as electrophiles that form adducts with thiols and amines. We have previously shown that 1,4-naphthoquinone (1,4-NQ) and its derivatives (NQs) catalytically oxidize H_2_S to inorganic hydropolysulfides, thiosulfate, and related sulfoxides [16]. Here we show that these inorganic products of NQ-catalyzed H_2_S oxidation may then react with other small thiols to form biologically relevant organic hydropolysulfides and sulfoxides. We also show that NQ-thiol adducts specifically affect the catalytic efficacy of NQs and that these reactions depend on both the specific structure of the NQ and the nature of the thiol involved.

### 3.1. Production of Thiol Hydropersulfides, Hydropolysulfides and Sulfoxides

Inorganic hydroper- and hydropolysulfides could be initially produced by several mechanisms: (1) two one-electron oxidation reactions with the carbonyl oxygen that form two hydrosulfide radical intermediates that then combine to produce hydropersulfide (H_2_S_2_), (2) a single two-electron reaction that forms elemental sulfur S^0^ which then combines with H_2_S to form H_2_S_2_ and (3) or by consecutive Michael addition reactions of two H_2_S on a quinoid carbon that releases H_2_S_2_. These possible mechanisms are under current investigation. Regardless, the present studies show that organic hydroper- and hydropolysulfides can be derived from sulfur exchange between the organic thiol and H_2_S_2_ (Equation (1)) or direct addition of elemental sulfur to the organic thiol (Equation (2)). Note: while it is recognized that the pKas become progressively lower with increasing sulfur atoms, for simplicity, the sulfur reactants and products are shown in the fully protonated form.
H_2_S_2_ + GSH/Cys –> GSH/CysS_2_H + H_2_S (1)
S^0^ + GSH/Cys –> GSH/CysS_2_H(2)

Continuation of these reactions can produce organic hydropolysulfides with as many as four sulfur atoms, as shown in Figure 1. We did not observe any GSH-S_n_-GSH or Cys-S_n_-Cys (*n* = 2–6) polysulfides with LCMS, most likely because these will react with excess H_2_S to give the thiol and the hydroper- and hydropolysulfides, where R=GSH or Cys (Equation (3)),
H_2_S + RS_n_R –> RSH + RS_n_H(3)

A small amount of GSHS_4_ and GS_2_OH was also observed at 2 min and GSOH was present at 2 and 10 min. These oxygenated products appear to be due to reactions shown in Equations (4)–(7), as described by Winterbourn [23].
O_2_^•−^ + GSH –> GSO^•^ + OH^−^(4)
GSO^•^ + GSH –> GSOH + GS^•^(5)
and this becomes part of a chain reaction,
GS^•^ + GS^−^ –> GSSG^•−^(6)
followed by,
GSSG^•−^ + O_2_ –> GSSG + O_2_^•−^(7)
to regenerate superoxide and continue the chain. SOD would be expected to inhibit these reactions, which is what we observed (Appendix A).

### 3.2. Production of NQ-Thiol and Propylamine Adducts

#### 3.2.1. 1,4-NQ-GSH/Cys Adducts

Reduction of 1,4-NQ with NaBH_4_ results in a semiquinone EPR spectrum that could be the result of comproportionation between the residual oxidized 1,4-NQ and the newly formed, reduced 1,4-NQH_2_, or a one-electron oxidation of the reduced 1,4-NQH_2_ with oxygen as the electron acceptor. The fact that these semiquinones could be formed in the absence, or near absence of oxygen (Appendix A) indicates that comproportionation is probably a major factor in the formation of the semiquinones. This finding, coupled with the fact that SOD does not have much effect on the rate or extent of the oxygen consumption of NQ with GSH, one can infer that at large molar excess of NQ over GSH (4:1), oxygen consumption (reaction of the semiquinone with oxygen to make superoxide), and, therefore, superoxide formation is slow and that GSH reduction of NQ is fast. Thus, there is little GSH to react with superoxide and make a chain reaction. In either case, the fully oxidized NQ will be produced by a second oxygen-dependent reaction. Similarities in EPR spectra of 1,4-NQ reactions with low concentrations of either GSH or Cys (Figure 4) suggest that both GSH and Cys also reduce 1,4-NQ while forming the thiol adduct (Equation (8)):1,4-NQ + GSH(Cys) + H^+^ –> 1,4-NQH_2_-GS(Cys)(8)

Although the slightly altered spectrum may reflect a small population of the NQ-thiol semiquinone, it appears that most of the semiquinone is not one of a thiol adduct. This suggests that the reduced quinone-thiol adduct readily reduces the unreacted, oxidized quinone (Equation (9)):1,4-NQ + 1,4-NQH_2_-GS(Cys) –> 1,4-NQ-GS(Cys) + 1,4-NQH_2_(9)

There is an interesting loss of EPR signal when the thiol:1,4-NQ ratio is 0.5:1. We propose that at this ratio, half of the initial 1,4-NQ will be reduced to the 1,4-NQH_2_-GS/Cys adduct, and half will remain as the oxidized 1,4-NQ. Since all the oxygen will have been consumed (the oxygen concentration in buffer is <265 μM, only 10% of the thiol concentration), comproportionation will have to account for most of the semiquinone formation. If 1,4-NQ and 1,4-NQH_2_-GS(Cys) adducts cannot comproportionate, or do so very slowly, then there will be no semiquinone formation and no EPR signal. This is consistent with what we observed. It is also possible that the reduced 1,4-NQH_2_-GS(Cys) completely reduces the oxidized 1,4-NQ producing equal amounts of 1,4-NQH_2_ and 1,4-NQ-GS(Cys), again hindering comproportionation. However, this scenario is less likely as this reaction appears to require enzymatic catalysis [24].

We also found that with considerably lower concentrations of 1,4-NQ (10–30 μM), some, but not all, of the oxygen was readily consumed upon addition of either GSH or Cys (Figure 3 and Figure 5). Furthermore, the amount of oxygen consumed was essentially equivalent to the amount of 1,4-NQ-thiol adduct expected to be produced, regardless of whether 1,4-NQ or thiol was in excess (Figure 5). This suggests that this is a stoichiometric reaction and not a catalytic process. We also observed that the formation of the 1,4-NQH_2_-Cys adduct consistently consumed slightly more oxygen than formation of the 1,4-NQH_2_-GSH adduct. We interpret this as a two-step process where a small fraction of the 1,4-NQH_2_-Cys is autoxidized to 1,4-NQ-Cys but then reacts with a second Cys to again reduce the adduct, this time to 1,4-NQH_2_-2Cys which is also autoxidized. This is consistent with our LCMS observations of only one GSH forming an adduct with 1,4-NQ, whereas both one and to a lesser extent, two Cys adducts were observed (Figure 4F,G).

#### 3.2.2. 1,4-NQ-Propylamine Adducts

Our oxygen consumption experiments suggested that 1,4-NQ-propylamine adducts were slower to form compared to GSH or Cys and required elevated ratios of propylamine to 1,4-NQ. This was confirmed by our EPR studies that showed that while the 1,4-NQ-propylamine semiquinone spectrum was identical to the 1,4-NQ semiquinone, it did not appear until the propylamine:1,4-NQ ratio was 0.5:1. Furthermore, it did not become appreciably reduced until the ratio was 5:1 and was still evident at a 10:1 ratio (Figure 4D). These results are similar to what we observed with 1,4-NQ and either GSH or Cys, albeit at a considerably higher propylamine:1,4-NQ concentration ratio, and we attribute the difference to the amine being less reactive than the thiols. Grant et al. [25] observed the EPR signals of 1,4-NQ adducts with GSH, *N*-acetylcysteine and glycine. They also needed a considerably higher ratio of glycine to 1,4-NQ than *N*-acetylcysteine to 1,4-NQ (20:1 versus 0.25:1, respectively) to achieve comparable results and they also concluded that thiols were considerably more reactive than amines.

Our studies also suggested that once formed, the 1,4-NQ-propylamine adducts are relatively impervious to attack from either GSH or Cys. We do not know if propylamine can be displaced from 1,4-NQ by H_2_S; however, the observations that, in most cases, propylamine neither increased nor decreased polysulfide production when H_2_S was added to a variety of NQs suggest that thiol adducts (especially GSH adducts), but not propylamine adducts, specifically affect NQ catalyzed H_2_S oxidation.

#### 3.2.3. Adducts of Other NQs with GSH/Cys

The presence of side groups clearly and specifically affect adduct formation. Reduction of either juglone or plumbagin with a low concentration of NaBH_4_ produced semiquinones that were considerably more complex than the 1,4-NQ semiquinone (cf. Figure 4 and Appendix A). The radicals formed at the low (0.05:1) GSH or Cys:juglone ratio is dominated by the juglone semiquinone radical, although they also exhibit some features of another radical. If the experiment is performed in anoxia, the juglone semiquinone radical disappears when the GSH:juglone ratio exceeds 0.5:1, whereas in 21% oxygen the previously faint radical becomes the dominant species, possibly a 2GS-juglone species, and it persists, albeit progressively fainter until the ratio exceeds 1:1. This radical was not as obvious with Cys even in 21% oxygen.

The radical observed with plumbagin and GSH does not appear to be the plumbagin semiquinone radical but probably that of a plumbagin-GS semiquinone. Also, GSH appeared to react with greater difficulty with plumbagin than it did with juglone, as suggested by the persistent observation of significant radical formation at a GSH:plumbagin ratio of 1:1. This is reminiscent of the persistence of an EPR signal we observed at high propylamine:1,4-NQ ratios (Figure 4D). Inbaraj and Chignell [26] incubated HaCaT Keratinocytes with plumbagin or juglone and observed that while plumbagin stoichiometrically converted GSH to GSSG, suggesting a redox cycling pathway, juglone appeared to preferentially form a juglone-GSH adduct.

Although a semiquinone radical is formed when 2-MNQ is reduced with NaBH_4_, no radicals were evident with either GSH or Cys (Appendix A). The decrease in propensity for semiquinone radical formation with GSH or Cys as the side chain complexity of the NQ increased is also observed as a decrease in oxygen consumption during adduct formation with more complex NQs (Appendix A).

### 3.3. Effects of GSH, Cys and Propylamine Adducts on H_2_S Oxidation

Incubating relatively high concentrations of NQs with equal concentrations of GSH, Cys, propylamine, and H_2_S for an extended period in air-equilibrated conditions before diluting them to low-micromolar concentrations allowed us to evaluate the effects of oxidized NQ adducts on H_2_S oxidation, we assume with relatively little interference from unreacted compounds (Figure 6 and Figure 7 and Appendix A). Consistent with other experiments, 10 μM NQs were generally more efficacious than 30 μM NQs and only 10 μM adducts will be considered.

With a 1:1 ratio of GSH:NQ, all NQ-GSH adducts except 2-MNQ increased polysulfide production (SSP4 fluorescence) when incubated with H_2_S. The order of potency (H_2_S + NQ + GSH fluorescence relative to H_2_S + NQ fluorescence) was menadione > plumbagin >> 1,4-NQ~juglone with 2-MNQ having no effect. Incubation of 10 μM of the menadione-GSH adduct with 100 μM H_2_S increased fluorescence over 13 times. Polysulfide production from H_2_S was also positively correlated with the GSH:NQ ratio and production significantly increased when this ratio was as low as 0.063:1 (juglone and plumbagin), 0.125:1 (menadione) or 0.25:1 (1,4-NQ). Polysulfide production was also increased by H_2_S incubation with 0.25:1 ratio of GSH:2-MNQ but declined as the ratio increased.

Conversely, 2-MNQ-Cys adducts were far more efficacious in H_2_S oxidation, followed by 1,4-NQ. Menadione-Cys adducts had only a slight effect, juglone and plumbagin adducts were ineffective and 30 μM plumbagin-Cys decreased polysulfide production four-fold. With the exception of 2-MNQ-Cys, NQ-Cys adducts were generally less sensitive to the Cys:NQ concentration ratio.

Although propylamine forms adducts with NQs and consumes oxygen in the process (Appendix A), it had minimal effect on H_2_S oxidation to polysulfides. Formation of a 1,4-NQ-propylamine adduct was slower than either the 1,4-NQ-GSH or 1,4-NQ-Cys adduct, and either thiol would out-compete propylamine if all were added within 15 min of each other. However, once a propylamine adduct was formed it appeared to prevent GSH from forming an adduct with NQ.

### 3.4. NQ-H_2_S Adducts

H_2_S-NQ reactions are unique in that H_2_S may not only form an adduct with the NQ, but some of the products of H_2_S oxidation may also form adducts. This is supported by our observations that the EPR spectra from H_2_S-NQ reactions were unique to this thiol (cf. Figure 4 and Appendix A). While addition of 600 μM H_2_S to 4 mM 1,4-NQ, juglone, or plumbagin (0.15:1 H_2_S:NQ) produced a spectrum that was essentially identical to the semiquinone produced by reduction of the NQ with NaBH_4_, and similar to those observed with GSH and Cys, the NQ-H_2_S spectrum did not disappear when the H_2_S:NQ ratio increased above 0.5:1, and now it no longer resembled the original semiquinone. These adducts could be due to reactions of NQ with a variety of inorganic per- and polysulfides or even sulfoxides. Ogata et al. [27] demonstrated that thiosulfate forms an adduct with benzoquinone in aqueous solutions. How, or if these other adducts affect H_2_S oxidation remains to be determined.

### 3.5. Autoxidation of NQ Adducts and Their Parent Compounds

Redox cycling is implicit in the NQ-catalyzed oxidation of H_2_S. While autoxidation reactions may affect the rate and nature of sulfur metabolism, they were not the primary goal of this study. Buffington et al. [20] compared autoxidation rate (H_2_O_2_ production) of parent NQs and their GSH conjugates after reduction by DT-diaphorase and NADPH and they reported that the glutathionyl conjugates “autoxidize at rates higher than those for the unsubstituted parent compounds”. Indeed, they showed that H_2_O_2_ production by 3-GS-1,4-NQ was twenty times greater than that produced by 1,4-NQ and 3-GS-menadione produced ten times more H_2_O_2_ than menadione. However, they also found that H_2_O_2_ production by juglone was ten times greater than that produced by 3-GS-juglone, and plumbagin produced five times as much H_2_O_2_ as 3-GS-plumbagin [20]. Moreover, they observed that autoxidation of either juglone or plumbagin consumed fifty times as much H_2_O_2_ as autoxidation of 1,4-NQ. Clearly these values are not reconcilable with our studies, and they support our hypotheses that NQ oxidation of H_2_S is a unique process. Furthermore, relevant autoxidation processes and mechanisms need to be considered in future work in this field.

## 4. Materials and Methods

### 4.1. H_2_S and Polysulfide Measurements in Buffer

Fluorophore experiments were performed in 96-well plates and fluorescence was measured with a SpectraMax M5e plate reader (Molecular Devices, Sunnyvale, CA, USA). Compounds were pipetted into 96-well plates and the plates were covered with tape to minimize H_2_S loss due to volatilization. Excitation/emission (Ex/Em) wavelengths were per the manufacturer’s recommendations. 7-azido-4-methylcoumarin (AzMC, 365/450 nm) and 3′,6′-Di(O-thiosalicyl) fluorescein (SSP4, 482/515 nm) have been shown to have sufficient specificity relative to other sulfur compounds and reactive oxygen and nitrogen species (ROS and RNS, respectively) to effectively identify H_2_S (AzMC) and per- and hydropolysulfides (H_2_S_2_ and H_2_S_n_ where *n* = 3–7 or RS_n_H where *n* > 1 or RS_n_R = where *n* > 2) [28,29,30]. As both AzMC and SSP4 are irreversible, they provide a cumulative record of H_2_S and polysulfide production, but they do not reflect cellular concentrations at any specific time.

### 4.2. Mass Spectrometry

Inorganic polysulfides were identified using liquid chromatography electrospray ionization high-resolution mass spectrometry (LCMS) using a micrOTOF-Q II Mass Spectrometer (Bruker Daltronics, Billerica, MA, USA) coupled to an UltiMate 3000 (Thermo Fisher, Waltham, MA, USA) UHPLC system, as described previously [18]. A Waters Acquity UPLC HSS T3 column (1.8 µm, 150 mm × 2.1 mm inner diameter; Waters Corporation, Wood Dale, IL, USA) was used with mobile phases A (water containing 0.1% formic acid) and B (acetonitrile containing 0.1% formic acid). Samples were diluted 10-fold in water and 20 µL was injected with a linear gradient (0–90% B, 30 min) at a flow rate of 0.4 mL/min. A sample of K_2_S_x_ (mixture of polysulfides) was derivatized with 1 M tyrosine methylester-iodoacetamide (TME-IAM) and used to determine the chromatographic behavior of the adducts. The mass spectrometer was used in the positive ion mode with the capillary voltage set to 2200 V and drying gas set to 8.0 L/min at 180 °C. The TME-IAM polysulfide adducts were detected as the [M + Na]^+^ ion using their exact masses ±0.002 *m*/*z*: S_1_ (505.164, 527.146), S_2_ (537.136, 527.146), S_3_ (569.108, 591.090), S_4_ (601.080, 623.062), CysS_1_ (357.111, 379.093), CysS_2_ (389.083, 411.066), CysS_3_ (421.055, 443.038), CysS_4_ (453.028, 475.010), Cys(S_2_)Cys (241.031, 263.013), Cys(S_3_)Cys (273.003, 294.985), Cys(S_4_)Cys (304.975, 379.093), GSHS_1_ (543.175, 565.158), GSHS_2_ (575.147, 597.130), GSHS_3_ (607.120, 629.102), GSHS_4_ (639.092, 661.074), GSH(S_2_)GSH (613.159, 635.141), GSH(S_3_)GSH (645.131, 667.113), and GSG(S_4_)GSH (677.103, 699.085).

The derivatizing agent TME-IAM (methyl (2-iodoacetyl)-L-tyrosinate) was synthesized in a one-step carbodiimide coupling reaction, as described previously [31], with modifications as described here. Iodoacetic acid (0.36 g, 1.9 mmol), L-tyrosine methyl ester (0.40 g, 2.1 mmol), and *N*,*N*′-dicyclohexylcarbodiimide (0.90 g, 4.4 mmol) were dissolved in 4 mL of dimethylformamide. The reaction mixture was stirred on ice and allowed to warm to room temperature over 3 h. Ethyl acetate (20 mL) was added, and the solid precipitate was removed by vacuum filtration. The filtrate was transferred to a separatory funnel and washed with water (4 × 10 mL). The organic layer was dried over anhydrous Na_2_SO_4_ and concentrated by rotary evaporation. To further remove urea byproduct, the residue was dissolved in cold acetonitrile (3 mL) and placed in the freezer for 30 min. Crude TME-IAM dissolved in acetonitrile was decanted, concentrated, and purified by automated flash column chromatography (Biotage Isolera, 25g Sfär Silica column, hexane/ethyl acetate gradient). Upon evaporation and lyophilization, 0.022 g (3% yield) of TME-IAM was obtained as a white solid. ^1^H NMR (500 MHz, CDCl_3_): δ ppm 6.99 (2 H, d, *J* = 8.2 Hz) 6.76 (2 H, d, *J* = 8.2 Hz) 6.47 (1 H, d, *J* = 7.3 Hz) 5.53 (1 H, br. s.) 4.83 (1 H, m) 3.77 (3 H, s) 3.68 (2 H, m) 3.08 (2 H, m); LCMS (ESI-TOF): >95% purity, [M + H]^+^ calculated *m*/*z* for C_12_H_15_INO_4_ 364.0040, found 364.0033.

In a typical experiment, H_2_S (as Na_2_S) was incubated in 10 mM phosphate buffer with NQ or other compounds of interest and aliquots removed at approximately 2, 10, 30, and 60 min and derivatized with TME-IAM. After 30 min, the samples were diluted 1:10 in 0.1% formic acid to stop the reaction and subjected to LCMS within 1 h. In initial studies, we noticed that there was some time-dependent loss in activity between the first and last samples that were subjected to LCMS. To monitor this, a second aliquot of the sample derivatized at 2 min was subjected to LCMS after the 60 min sample for comparison.

### 4.3. Oxygen Consumption by Naphthoquinones, H_2_S and Thiol/Amine Adducts

Oxygen tension was monitored in a stirred 1 mL water-jacketed chamber with a FireStingO_2_ oxygen sensing system (Pyroscience Sensor Technology, Aachen, Germany) using a non-oxygen consuming 3 mm diameter OXROB10 fiberoptic probe at room temperature. The probe was calibrated with room air (21% O_2_) or nitrogen gas (0% O_2_). Compounds of interest were added at timed intervals and percent oxygen (100% equals room air) was measured every 0.3 s. Oxygen concentration in μM was estimated by multiplying the percent oxygen by the solubility coefficient for oxygen in 300 mOsm saline at 20 °C (2.65 μM∙L^−1^∙% O_2_^−1^, i.e., for air saturated buffer, 2.65 × 100 = 265 μM oxygen).

### 4.4. Thiosulfate Production

Thiosulfate was measured using silver nanoparticles (AgNP) as previously described [32]. Briefly, AgNPs were prepared by reducing AgNO_3_ with tannic acid (TA) in the presence of HAuCl_4_·4H_2_O. A total of 1 mL of 20 mM AgNO_3_ was mixed with 200 μL of 0.5 mM HAuCl_4_ in 98 mL of Milli-Q water at room temperature. A total of 1 mL of 5.0 mM TA was added, and the mixture was vigorously stirred, turning yellow within 30 min. The AgNPs were stored at 4 °C until use. Thiosulfate standards were made fresh daily, and a standard curve was prepared by plotting the concentration against (A_0_ − A)/A_0_, where A_0_ and A are absorbances of AgNPs without and with thiosulfate, respectively. Dong et al. [32] reported a linear standard curve; however, we note that the standard curve was non-linear but reproducible.

In a typical experiment, H_2_S, 1,4-NQ, and either GSH or Cys were placed in 96-well plates and the plates were covered with tape for 60 min to minimize H_2_S volatilization during the reaction, then the tape was removed to allow excess H_2_S to dissipate for an additional 60 min as we noted some H_2_S interference with this assay. A total of 30 μL of the buffer was then added to 200 μL of the AgNP in 96-well plates and the absorbance was measured after 60 min at 419 nm.

Thiosulfate was also measured using a variation of the high-performance liquid chromatography-monobromobimane (HPLC-MBB) assay reported previously [33,34]. HPLC-fluorescence analysis was performed with an UltiMate 3000 HPLC (Thermo Fisher Scientific, Waltham, MA, USA) with fluorescence detection (Ultimate FLD-3100 fluorescence detector, Thermo Fisher Scientific). A Shim-pack VP-ODS column (5 μm, 4.6 × 150 mm, Shimadzu Scientific, Kyoto, Japan) was used with mobile phases A (MilliQ water containing 0.25% formic acid) and B (methanol containing 0.25% formic acid). Samples (10 μL) were injected onto the column and run with a linear gradient (0–75% B) at a flow rate of 0.8 mL/min and a column temperature of 30 °C. The detector excitation/emission was set to 370/485 nm. Thiosulfate concentration was derived from the standard curve prepared on the same day. In these experiments, 50 μL of Na_2_S (60 μM), 1,4-NQ (6 μM), and either GSH (200 μM) or Cys (200 μM), all final concentrations, were added to Eppendorf tubes; the tubes were then closed for 70 min to allow the reaction to proceed. The tubes were then opened for an additional 60 min to allow H_2_S volatilization and 10 μL of 25 mM MBB in acetonitrile was added and the samples incubated in the dark at room temperature for 30 min to convert the thiosulfate into thiosulfatebimane.

### 4.5. Absorbance Spectra

Absorbance spectra were measured with an Agilent HP 8453 spectrometer (Agilent Technologies, Santa Clara, CA, USA). Experiments were performed at room temperature with PBS equilibrated with room air (21% O_2_) or PBS sparged for 30 min with nitrogen (0% O_2_) and spectra from 190 to 1100 nm were obtained at 4–6 s intervals for at least 25 min. In a typical experiment, the naphthoquinone was added, and three consecutive spectra obtained before addition of other reagents. Figures are displayed over 190–500 nm for clarity as most of the effects on absorbance were observed over this range.

### 4.6. Electron Paramagnetic Resonance (EPR) Spectrometry

X-band EPR spectra were recorded at 295 K on a Bruker EMX spectrometer (Billerica, MA, USA). The instrument parameters were frequency, 9.30 GHz; MW power, 1 or 4 mW; range, 20 G; modulation frequency, 100 kHz; modulation amplitude, 0.1 or 0.2 G; and time constant, 0.17 s.

EPR samples were prepared by diluting 16 mM 1,4-NQ DMSO stocks in PBS buffer (pH 7.4, final 1,4-NQ concentration 4 mM) and reacted with GSH, Cys, or H_2_S prepared in PBS buffer. H_2_S stock was prepared by dissolving Na_2_S in PBS. For hypoxia reactions, 1,4-NQs were diluted into N_2_-sparged PBS buffer and the solutions were further sparged with N_2_ while H_2_S was prepared by dissolving Na_2_S into N_2_-sparged PBS. Reaction solutions were then transferred into capillary tubes for EPR measurements. To prepare 1,4-NQ semiquinone radical, 4 mM 1,4-NQs were reacted with 0.15–0.25× sodium borohydride (NaBH_4_).

The simulation of EPR spectra was conducted using Simfonia (Bruker). The parameters used in the simulation were g = 2.0044 and hyperfine splitting of 3.21 G for protons 2 and 3 and 0.63 G for protons 5, 6, 7, and 8. The parameters for simulating the spectrum of the SH-substituted 1,4-NQ were g = 2.0044 and hyperfine splitting of 0.69 G for protons 5, 6, 7, and 8.

### 4.7. H_2_S Oxidation and Polysulfide Production by 1,4-NQ-Thiol and 1,4-NQ-Amine Adducts

Since thiols interfere with the SSP4 detection of polysulfides from H_2_S/NQ reactions (see Results, Section 2.1), we prepared concentrated adducts and then diluted them prior to adding H_2_S and SSP4. One mM of GSH, Cys, or propylamine (PA) was added to 1 mM 1,4-NQ and allowed to react for 1–2 h in an open vessel to form the adduct and then allow it to autoxidize. We also assumed that most if not all the thiol or PA would be consumed in this reaction. The aliquots were then diluted in buffer in 96-well plates to a final concentration of 10 μM or 30 μM, leaving little or no free thiol or PA. H_2_S-NQ adducts were initially reacted for 30 min in a closed container to minimize H_2_S volatilization; the containers were then opened for another hour to allow all the remaining free H_2_S to dissipate and to oxidize the NQ-H_2_S adduct. SSP4 and H_2_S were then added, and fluorescence was measured over 90 min and compared to control.

### 4.8. Chemicals

SSP4 was purchased from Dojindo molecular Technologies Inc. (Rockville, MD, USA). All other chemicals were purchased from Sigma-Aldrich (St. Louis, MO, USA) or ThermoFisher Scientific (Grand Island, NY, USA). H_2_S is used throughout to denote the total sulfide (sum of H_2_S + HS^−^) derived from Na_2_S, as S^2−^ most likely does not exist under these conditions [35]. Phosphate buffered saline (PBS; in mM): 137 NaCl, 2.7 KCl, 8 Na_2_HPO_4_, and 2 NaH_2_PO_4_. Phosphate buffer for absorbance measurements (PB; in mM): 200 Na_2_PO_4_. pH was adjusted with 10 mM HCl or NaOH to pH 7.4.

### 4.9. Statistical Analysis

Data were analyzed and graphed using QuatroPro (Corel Corporation, Ottawa, ON, Canada) and SigmaPlot 13.0 (Systat Software, Inc., San Jose, CA, USA). Statistical significance was determined with Student’s t-test or one-way ANOVA and the Holm–Sidak test for multiple comparisons as appropriate using SigmaStat (Systat Software, San Jose, CA, USA). Results are given as mean ± SE; significance was assumed when *p* < 0.05.

## 5. Conclusions

We previously demonstrated that a variety of NQs catalytically oxidize H_2_S to produce inorganic hydroper- and hydropolysulfides and sulfoxides. Here, we show that NQ-mediated H_2_S oxidization in the presence of GSH and Cys produces organic as well as inorganic hydroper- and hydropolysulfides and sulfoxides. NQs also form adducts with these thiols and protein Cys, and this may augment or decrease H_2_S oxidation and the nature of product production. These processes are especially germane given the prevalence of GSH and protein Cys in the cytosol. Organic and inorganic hydroper- and hydropolysulfides are highly efficient ROS scavengers; they can increase the incorporation of sulfane sulfur into small mobile organic persulfides for storage or signaling, and they may affect the activity of regulatory proteins. Furthermore, both hydroper- and hydropolysulfides and NQs can liberate Keap-1 from Nrf2, thereby activating the Keap-1/Nrf2 antioxidant system. While the antioxidant and pro-oxidant functions of NQs have been largely attributed to the effects of NQs on ROS, the effects of NQs on reactive sulfur species (RSS) may be even more significant, especially in the low-oxygen environments of most cells.

## Figures and Tables

**Figure 1 ijms-24-07516-f001:**
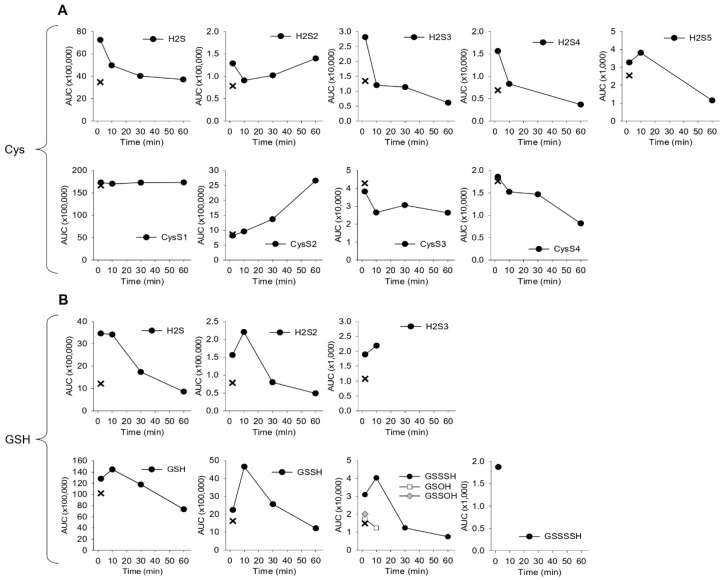
Mass spectrometric evaluation of inorganic and organic sulfur compounds produced by 10 μM 1,4-NQ oxidation of 300 μM H_2_S in the presence of either 1 mM cysteine ((**A**); Cys) or 1 mM glutathione ((**B**); GSH). Values are expressed as relative area under the curve (AUC). GSH or Cys were added to 1,4-NQ followed by H_2_S. TME-IAM was added at 2, 10, 30, and 60 min and the reactions were stopped with 1% formic acid 30 min after the last (60 min) sample and subjected to mass spectrometry. A second aliquot of the 2 min sample was run after the 60 min sample (**✖** in figures) to evaluate the stability of the TME-IAM-sulfur adducts.

**Figure 2 ijms-24-07516-f002:**
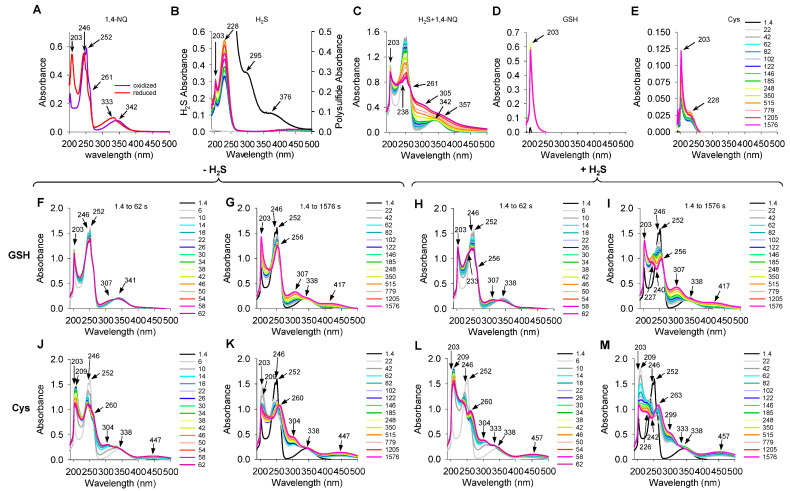
Absorption spectra of 1,4-NQ and reactions with thiols. (**A**–**D**) Reference spectra of oxidized and N_2_-reduced 1,4-NQ (**A**), 60 μM H_2_S and polysulfides (**B**), 60 μM H_2_S plus 60 μM 1,4-NQ (**C**), GSH (**D**), and Cys (**E**). Time-resolved absorption spectra of 1,4-NQ (60 μM) and GSH (60 μM; (**F**–**I**)) or Cys (60 μM; (**J**–**M**)) without (-H_2_S; (**F**,**G**,**J**,**K**)) or with (**H**,**I**,**L**,**M**) 60 μM H_2_S. (**F**,**H**,**J**,**L**) show responses over the initial 62 s and (**G**,**I**,**K**,**M**) show responses over 1576 s. Time key at the right indicates elapsed time in seconds and arrows show wavelengths of major peaks.

**Figure 3 ijms-24-07516-f003:**
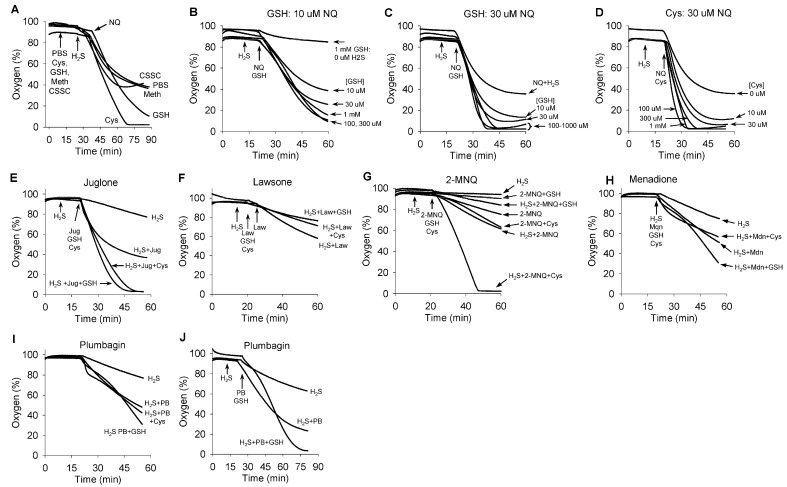
Typical traces of the effects of low-molecular-weight thiols on O_2_ consumption by H_2_S and naphthoquinones under various conditions. (**A**) O_2_ consumption by 300 μM H_2_S and 10 μM 1,4-NQ (NQ) alone (PBS) or in the presence of cysteine (Cys, 1 mM), glutathione (GSH, 1 mM), cystine (CSSC, 1 mM), and methionine (Meth, 1 mM). Cys and GSH increase O_2_ consumption, CSSC and Meth do not. (**B**,**C**) GSH concentration-dependently increases O_2_ consumption by 300 μM H_2_S and 10 μM 1,4-NQ (**B**) or 30 μM 1,4-NQ (**C**). (**D**) Cys concentration-dependently increases O_2_ consumption by 300 μM H_2_S and 30 μM 1,4-NQ. (**E**) O_2_ consumption by 30 μM juglone (Jug) and 300 μM H_2_S is increased by both 1 mM GSH and 1 mM Cys. (**F**) O_2_ consumption by 30 μM lawsone and 300 μM H_2_S is minimal and slightly decreased by both 1 mM Cys and 1 mM GSH. (**G**) Cys (1 mM) greatly increases O_2_ consumption by 10 μM 2-MNQ and 300 μM H_2_S, whereas 1 mM GSH decreases it. (**H**) GSH (1 mM) increases O_2_ consumption by 30 μM menadione (Mdn) and 300 μM H_2_S, whereas it is unaffected by 1 mM Cys. (**I**) Plumbagin (PB; 30 μM) increases O_2_ consumption, 1 mM Cys does not affect it, and 1 mM GSH further increases it after a slight delay. (**J**) Expanded timescale showing delay and then increased O_2_ by addition of GSH to 30 μM plumbagin and 300 μM H_2_S.

**Figure 4 ijms-24-07516-f004:**
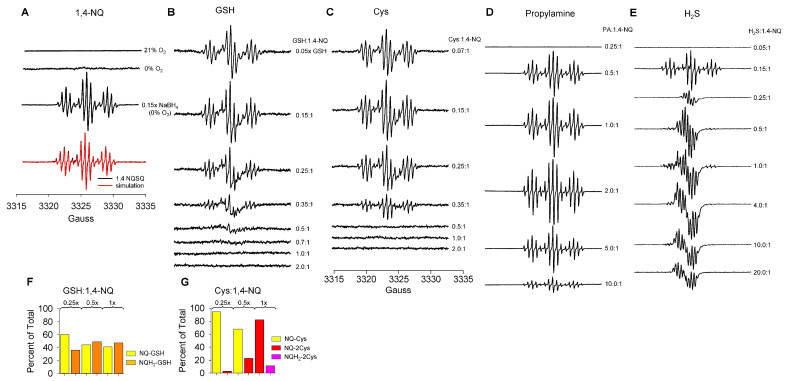
1,4-NQ-GSH and 1,4-NQ-Cys adducts detected by EPR (**A**–**G**) and LCMS (**F**,**G**). (**A**) EPR spectra of 4 mM 1,4-NQ in 21% and 0% O_2_ is flat, whereas reduction of 4 mM 1,4-NQ with 1 mM NaBH_4_ and subsequent comproportionation produces the 1,4-NQ semiquinone that is identical to the simulated semiquinone (red trace). (**B**–**E**) EPR spectra of 4 mM 1,4-NQ with increasing concentrations (shown as the ratio of thiol to 1,4-NQ) of GSH (**B**), Cys (**C**), propylamine (**D**), or H_2_S (**E**–**G**). Spectra of GSH and Cys are similar but not identical to the 1,4-NQ semiquinone at low GSH or Cys concentrations, and the signals disappear as the ratio of GSH or Cys to 1,4-NQ approaches 0.5:1. Spectra of propylamine are identical to reduced 1,4-NQ semiquinone and persist up to a 10:1 propylamine:1,4-NQ ratio. EPR spectra of 4 mM 1,4-NQ and increasing concentrations of H_2_S are different. At a H_2_S:1,4-NQ ratio of 0.15:1, the EPR spectrum is also typical of the 1,4-NQ semiquinone, whereas the spectrum abruptly changes and becomes more complex at higher H_2_S:1,4-NQ ratios. These changes persist even at a H_2_S:1,4-NQ ratio of 20:1. (**F**) Incubation of 250 μM, 500 μM, and 1 mM GSH with 1 mM 1,4-NQ (0.25:1, 0.5:1 and 10:1 GSH:NQ) produced nearly equal amounts of 1,4-NQ-GSH and 1,4-NQH_2_-GSH. (**G**) Incubation of 250 μM, 500 μM, and 1 mM Cys with 1 mM 1,4-NQ (0.25:1, 0.5:1 and 1.01 Cys:1,4-NQ) produced mostly 1,4-NQ-Cys at low Cys concentrations, whereas 1,4-NQ-2Cys predominate at 1:1 Cys:NQ. Little 1,4-NQH_2_-Cys was observed ((**F**,**G**); average of *n* = 2).

**Figure 5 ijms-24-07516-f005:**
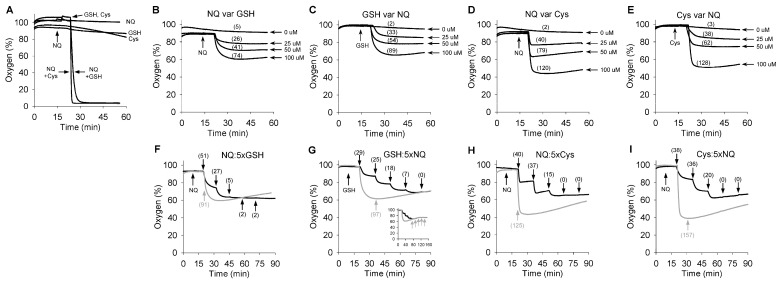
Oxygen consumption by mixtures of 1,4-NQ, GSH, and Cys. (**A**) Oxygen is rapidly consumed by 200 μM of GSH or 280 μM of Cys added to 4 mM 1,4-NQ (0.05:1 and 0.07:1 thiol:1,4-NQ, respectively), but neither thiol nor 1,4-NQ consumes oxygen alone. (**B**–**E**) Oxygen consumption by 100 μM 1,4-NQ and increasing thiol concentrations (**B**,**D**) or by 100 μM thiol with increasing 1,4-NQ concentrations (**C**,**E**). (**F**,**G**) Oxygen consumption following a single addition of 125 μM (gray line and symbols) or five sequential additions of 25 μM (black line and symbols) of GSH or Cys to 100 μM 1,4-NQ ((**F**,**H**), respectively), or single and sequential additions of 1,4-NQ to 100 μM GSH or Cys ((**G**,**I**), respectively). Expanded timescale in (**G**) (inset) shows the lack of effect of 5 additions of 25 μM 1,4-NQ (gray arrows) to the single 125 μM GSH-100 μM 1,4-NQ sample. Values in parentheses show the net decrease in oxygen concentration in μM for each treatment.

**Figure 6 ijms-24-07516-f006:**
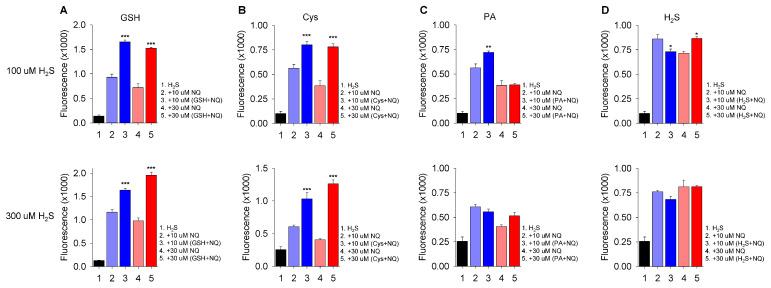
Polysulfide production (SSP4 fluorescence) from H_2_S catalyzed by 1,4-NQ adducts. Adducts were prepared by incubating 1 mM 1,4-NQ with 1 mM GSH (**A**), Cys (**B**), PA (**C**), or H_2_S (**D**) for 1 h then reacting 10 μM or 30 μM of the adduct with either 100 μM (top panels) or 300 μM (bottom panels) H_2_S. Polysulfide production was greatly increased by each thiol adduct but only marginally affected by the PA or H_2_S adducts; GSH adducts were the most efficacious. Values summarize effects after 90 min reaction, mean + SE, *n* = 4 wells; *, *p* < 0.05, **, *p* < 0.01, ***, *p* < 0.001 vs. 1,4-NQ without adduct.

**Figure 7 ijms-24-07516-f007:**
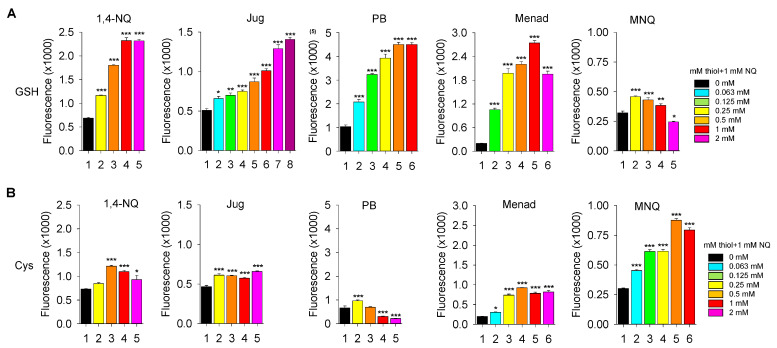
Effects of the ratio of GHS or Cys to NQ on polysulfide production (SSP4 fluorescence). A total of 1 mM of NQ was incubated with various concentrations of GSH (**A**) or Cys (**B**) for 1 h, diluted to 10 μM NQ, and incubated with 300 μM H_2_S and SSP4 for 90 min. Most GSH adducts concentration-dependently increased polysulfide production. Only menadione (Mdn) and 2-MNQ (MNQ) Cys adducts were concentration dependent. Values summarize effects after 90 min reaction with H_2_S, mean + SE, *n* = 4 wells; *, *p* < 0.05, **, *p* < 0.01, ***, *p* < 0.001 vs. NQ without thiol.

## Data Availability

Data are contained within the article or Appendix A.

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
