# Peer review of "Redox and Nucleophilic Reactions of Naphthoquinones with Small Thiols and Their Effects on Oxidization of H2S to Inorganic and Organic Hydropolysulfides and Thiosulfate"

_ijms, 2023, doi:10.3390/ijms24087516_

Round 1

Reviewer 1 Report

The manuscript by Olson and coworkers describes the effects of thiols and thiol-naphthoquinone (thiol-NQ) adducts on hydrogen sulfide-naphthoquinones (H2S-NQ) reactions. The authors have presented the reducing reactions mediated by naphthoquinones (NQ) and examined the progress by the oxygen consumption via a semiquinone intermediate formation. NQs are also reduced by the formation of adducts with thiols including protein thiols and amines. The authors described and discussed a variety of analyses from these interactions, including mass spectrometric evaluation, absorption spectrometry, oxygen consumption analysis, fluorescence, and others. Although the results are well substantiated, and the presented data are important to enrich the role of NQ-mediated oxidation process, I suggest the analysis and discussion of radical formation species. The present work, I consider as a contribution for the scientific community although it requires revisions.

Figure 1 could be better illustrated. The results presented are confusing and difficult to read and understand. I suggest the authors to select data and insert part of the data in the supporting information file.

In the paragraph about quinones, I suggest also expose the fact that quinones are a versatile and privileged structures regarding biological activities, please cite, in general (Eur. J. Med. Chem., 2019, 179, 863-915), not only as an anticancer agent (Toxicol. In Vitro, 2012, 26, 585-594; Molecules, 2018, 23:83; Eur. J. Med. Chem., 2018, 151, 686-704) but also as a tripanocidal (Eur. J. Med. Chem., 2017, 136, 406-419; Chem. Eur. J., 2018, 15227-15235), antifungal (PLoS One, 2014, 9, e93698), tuberculosis (Eur. J. Med. Chem., 2011, 46, 4521-4529) or even insecticidal (Genet. Mol. Biol., 2022, 45, e20210307) agent.

Authors considered TBARS assays?

Author Response

We thank reviewer 1 for the time and thoughtful, constructive criticisms.  These have been incorporated into the manuscript and we believe it has been considerably improved.

Reviewer 1                                                                             

The manuscript by Olson and coworkers describes the effects of thiols and thiol-naphthoquinone (thiol-NQ) adducts on hydrogen sulfide-naphthoquinones (H2S-NQ) reactions. The authors have presented the reducing reactions mediated by naphthoquinones (NQ) and examined the progress by the oxygen consumption via a semiquinone intermediate formation. NQs are also reduced by the formation of adducts with thiols including protein thiols and amines. The authors described and discussed a variety of analyses from these interactions, including mass spectrometric evaluation, absorption spectrometry, oxygen consumption analysis, fluorescence, and others. Although the results are well substantiated, and the presented data are important to enrich the role of NQ-mediated oxidation process, I suggest the analysis and discussion of radical formation species. The present work, I consider as a contribution for the scientific community although it requires revisions.

* We thank the reviewer for these comments and suggestions.  As the reviewer points out, formation of various radicals is is common to these reactions and it is a complex process.  We discussed some of the possible reactions and radicals involved but the primary intent of this study was to show that NQs react with H2S and biologically relevant thiols and that this can, 1) produce organic hydroper- and hydropolysulfides and that 2) NQ-thiol adducts can affect the catalytic properties of NQs.  As stated in the discussion, radical production during the autoxidation process was not the focus of this study as many additional studies (currently underway) are necessary to describe these reactions.  We beg the reviewer’s indulgence in delaying an extensive discussion of these reactions.

Figure 1 could be better illustrated. The results presented are confusing and difficult to read and understand. I suggest the authors to select data and insert part of the data in the supporting information file.

* Figure 1 has been redrawn and Figure 1C has been removed and placed in the supplemental figures as Supplemental Fig. 1.

In the paragraph about quinones, I suggest also expose the fact that quinones are a versatile and privileged structures regarding biological activities, please cite, in general (Eur. J. Med. Chem., 2019, 179, 863-915), not only as an anticancer agent (Toxicol. In Vitro, 2012, 26, 585-594; Molecules, 2018, 23:83; Eur. J. Med. Chem., 2018, 151, 686-704) but also as a tripanocidal (Eur. J. Med. Chem., 2017, 136, 406-419; Chem. Eur. J., 2018, 15227-15235), antifungal (PLoS One, 2014, 9, e93698), tuberculosis (Eur. J. Med. Chem., 2011, 46, 4521-4529) or even insecticidal (Genet. Mol. Biol., 2022, 45, e20210307) agent.

* We thank the reviewer for these valuable references on activities of various naphthoquinones, some of which we are looking into for future studies.  They have been incorporated into the manuscript.

Authors considered TBARS assays?

* We thought about a number of assays of this type, but our primary focus was on the various reactions between NQs and H2S.  These assays would certainly be appropriate for cell studies.

Reviewer 2 Report

This manuscript describes the NQ-mediated H2S oxidization in the presence of GSH and Cys. Characterization including mass spectroscopy, absorbance spectroscopy, and EPR are conducted. The reaction mechanism is discussed in depth. The paper is well-written, and the methods are sufficiently described. I have a few minor comments.  

1.       It would be helpful to include the application of H2S-NQ reactions and H2S-NQ reactions with  in the introduction section.

2.       Figure 1/2/3 shows the experimental result of H2S-NQ reaction with the presence of Cys or GSH. How does it compare to the reference group without Cys and GSH?

3.       Figure 1 (A) legends need bigger font. It is very difficult to read the compound names.

4.       Figure 4 caption says subfigure (H, I) but I don’t find H and I.

Author Response

We thank reviewer 2 for the time and thoughtful, constructive criticisms.  These have been incorporated into the manuscript and we believe it has been considerably improved.

Reviewer 2

This manuscript describes the NQ-mediated H2S oxidization in the presence of GSH and Cys. Characterization including mass spectroscopy, absorbance spectroscopy, and EPR are conducted. The reaction mechanism is discussed in depth. The paper is well-written, and the methods are sufficiently described. I have a few minor comments.

  1. It would be helpful to include the application of H2S-NQ reactions and H2S-NQ reactions with in the introduction section.

* There was something missing in the reviewer’s comment, so we are not completely sure want was wanted.  Nevertheless, in anticipation of this request we added the following to the beginning of the last paragraph.  “Given the ability of NQs to form adducts with thiols and amines, we hypothesize that this could affect the catalytic properties of NQ-H2S reactions.  These would be especially relevant in intracellular environments where there can be an abundance of reactive thiols and amines.”

  1. Figure 1/2/3 shows the experimental result of H2S-NQ reaction with the presence of Cys or GSH. How does it compare to the reference group without Cys and GSH?

* Regarding Fig. 1, we added the following sentence to the beginning of the paragraph “We have previously shown that 1,4-NQ oxidizes H2S S2-S6 polysulfides (Olson et al., 2022{full reference in MS}).  Here we show that similar reactions occur in the presence of small thiols.”  Regarding Fig. 2, we added a panel to the figure (Fig. 2C) showing the reaction of H2S with 1,4-NQ.  Regarding Fig. 3, the relevant H2S-NQ effects are in the figures.

  1. Figure 1 (A) legends need bigger font. It is very difficult to read the compound names.

* The font has been increased and the Figure has been redone for additional clarity, including moving panel C to the supplementary figures

  1. Figure 4 caption says subfigure (H, I) but I don’t find H and I.

* Sorry, that was a clerical error and should have been F and G.  Now changed.

Reviewer 3 Report

The authours describe the inorganic  products of NQ-catalyzed H2S oxidation. The products were then react  with other small thiols to form biologically relevant organic hydropolysulfides and sulfoxides. So they can measure the effects of thiols, and thiol-NQ adducts on H2S-NQ reactions. The behaviour of amines were also studied and compared with thiol adducts. It has been used in some spectroscopic methods such as LC MS. Although it is a comprehensive study, important parts that will attract the attention of the reader can be given in a more summary form. In the abstract part, the determination methods can be briefly attributed. It would be nice if the references were expanded a bit more. The article is suitable for publication after these revisions.

Author Response

We thank reviewer 3 for the time and thoughtful, constructive criticisms.  These have been incorporated into the manuscript and we believe it has been considerably improved.

Reviewer 3

The authours describe the inorganic  products of NQ-catalyzed H2S oxidation. The products were then react  with other small thiols to form biologically relevant organic hydropolysulfides and sulfoxides. So they can measure the effects of thiols, and thiol-NQ adducts on H2S-NQ reactions. The behaviour of amines were also studied and compared with thiol adducts. It has been used in some spectroscopic methods such as LC MS. Although it is a comprehensive study, important parts that will attract the attention of the reader can be given in a more summary form. In the abstract part, the determination methods can be briefly attributed. It would be nice if the references were expanded a bit more. The article is suitable for publication after these revisions.

*  The abstract has been modified to include the methods and additional references inserted (also per recommendation of reviewer 1).